# Shikonin as a WT1 Inhibitor Promotes Promyeloid Leukemia Cell Differentiation

**DOI:** 10.3390/molecules27238264

**Published:** 2022-11-26

**Authors:** Zhenzhen Guo, Luyao Sun, Haojie Xia, Shibin Tian, Mengyue Liu, Jiejie Hou, Jiahuan Li, Haihong Lin, Gangjun Du

**Affiliations:** 1Henan Province Engineering Research Center of High Value Utilization to Natural Medical Resource in Yellow River Basin, Pharmaceutical College, Henan University, Kaifeng 475004, China; guozhenzhenlcy@163.com (Z.G.); sly15517830952@163.com (L.S.); zdc122333gy@163.com (H.X.); shibin006@163.com (S.T.); lmyuuup@163.com (M.L.); houjie12342021@163.com (J.H.); 2School of Pharmacy and Chemical Engineering, Zhengzhou University of Industry Technology, Xinzheng 451100, China

**Keywords:** shikonin, leukemia, Wilms’ tumor gene 1 (WT1), CD34, cell differentiation

## Abstract

This study aims to observe the differentiating effect of shikonin on Wilms’ tumor 1 (WT1)-positive HL-60 cells and investigate the fate of the differentiated leukemia cells. WT1 overexpression unaffected cell viability but promoted resistance to H_2_O_2_-induced DNA injury and cell apoptosis. The binding of shikonin to the WT1 protein was confirmed by molecular docking and drug affinity reaction target stability (DARTS). Shikonin at the non-cytotoxic concentration could decrease the WT1 protein and simultaneously reduced the CD34 protein and increased the CD11b protein in a dose-dependent manner in normal HL-60 cells but not in WT1-overexpressed HL-60 cells. Shikonin unaffected HL-60 cell viability in 48 h. However, it lasted for 10 days; could attenuate cell proliferation, mitochondrial membrane potential (MMP), and self-renewal; prevent the cell cycle; promote cell apoptosis. In a mouse leukemia model, shikonin could decrease the WT1 protein to prevent leukemia development in a dose-dependent manner. In this study, we also confirmed preliminarily the protein–protein interactions between WT1 and CD34 in molecular docking and CO-IP assay. Our results suggest that: 1. shikonin can down-regulate the WT1 protein level for leukemia differentiation therapy, and 2. the interaction between WT1 and CD34 proteins may be responsible for granulocyte/monocyte immaturity in HL-60 cells.

## 1. Introduction

Acute myeloid leukemia (AML) is the most common type of leukemia, accounting for 30% of childhood leukemia and 80% of adult leukemia [1]. In the clinic, cytotoxic chemotherapeutic regimens using anti-proliferative drugs or apoptosis-inducing agents have been another “golden standard” for AML therapy apart from bone marrow transplantation [2]. Although having certain effects, these cytotoxic regimens have not improved the low five-year survival rates in AML patents [3]. Comfortingly, all trans-retinoic acid (ATRA) has initiated a new era—leukemia differentiation therapy, which can induce the terminal differentiation of acute promyeloid leukemia (APL) cells and may achieve five-year overall survival rates of approximately 85% in APL patients with the predicted poor prognosis [4,5]. Unfortunately, this therapeutic utility has only been successful in the minor AML subtypes [6]. The finding of novel differentiation agents instead of cytotoxic chemotherapeutic drugs is a great challenge, and the identified oncogenic proteins in AML samples are just able to provide the potential targets for novel differentiation agents [7].

Wilms’ tumor 1 (WT1) is a transcription regulator and has four DNA-binding zinc-finger isoforms generated by splicing 17 amino acid (1 A) residues of exons 5 and 3, and amino acid (KTS) residues of exon 9 in the carboxyl terminal [8]. Although it has been believed to act as an oncogene [9], WT1 has also been reported to be considered as a tumor suppressor in AML [10]. In some studies, WT1 overexpression at the RNA and the protein level was observed in 90% of AML patients, which correlates to poor outcome, therefore being believed to function as an oncogenic potential in AML [11]. It was also reported that WT1 was downregulated in AML patients when the disease was controlled but was upregulated again upon disease relapse [12]. However, up until now, WT1 displays a controversial and complex function in hematological malignancies, for example, in the HL-60 cell line, the elimination of WT1 by RNA interference (RNAi) or antisense oligonucleotide prevented cell proliferation and cycle progression in Watanabe’s study [13]; however, in Yao’s study, the same elimination of WT1 did not affect cell viability and even promoted cell growth [14]. A rational explanation for the WT1 contrary function is that WT1 exerts different roles in an isoform, cell subtype, and targeted promotor-dependent manner [15]. Therefore, the function of WT1 in AML still remains to be elucidated.

HL-60 cells naturally express high levels of WT1 and are a preferential model for studying leukemia differentiation [16]. WT1 expression downregulation was considered a differentiation-linked event in HL-60 cells [17]. It was believed that WT1 expression was post-transcriptionally regulated because granulocytic/monocytic differentiation of HL-60 cells only resulted in WT1 protein downregulation but unaffected RNA transcription [18]. Currently, there are no small-molecule drugs as WT1 inhibitors. Arnebia euchroma (Royle) Johnst (AEJ, Zicao) is traditional Chinese medicine (TCM) and has been used to treat many kinds of illnesses [19,20], particularly blood diseases such as leukemia [21]. Shikonin as one of the main efficacy components in Zicao could exert multiple biological functions and demonstrated a promising activity against many cancer cells both in vitro and in vivo [22,23]. Most importantly, shikonin displayed anti-leukemic properties in AML primary cell lines [24], and promoted p53 or p21 expression for cancer prevention [25], implying its potential function as a differentiation agent. Therefore, this study aims to observe the differentiating effect of shikonin on Wilms’ tumor 1 (WT1)-positive HL-60 cells and investigate the fate of the differentiated leukemia cells.

## 2. Results

### 2.1. WT1 Overexpression Promoted HL-60 Cell Resistance to H_2_O_2_

Although WT1 overexpression is a characteristic feature in most AML, it was argued that WT1 was an oncogene or a tumor suppressor [9,10]. In order to explore the role of WT1 in AML, WT1-overexpressed HL-60 cells were established by Lentivirus transfection. As shown in Figure 1, WT1 expression in HL-60O cells was more three times than that in HL-60 cells (Figure 1A). WT1 overexpression unaffected cell viability (Figure 1B) and doubling time (Figure 1C, 23.98 h (23.09–24.87 h) for HL-60O cells vs. 24.22 h (23.10–25.34 h) for HL-60 cells, *p* > 0.05) but promoted cell resistance to H_2_O_2_-induced DNA injury (Figure 1D) and cell apoptosis (Figure 1E), indicating an injury-preventing role of WT1 in a toxic environment.

WT1 is an important transcription factor in embryonic development and participates in the regulation of cell differentiation [26]. It is well known that leukemia is characterized by arrested differentiation. Therefore, we speculated that the WT1 gene may be a key target for regulating leukemia cell differentiation. CD34 is a biomarker of the most immature phenotype in clinical leukemia and its expression is progressively lost as the cells mature and differentiate [27]. To prove this opinion, we examined CD34 expression and cell differentiation markers. It was shown that CD34 expression was upregulated in WT1-overexpressed HL-60 cells (Figure 2A), suggesting a role of WT1 in the immature phenotype in HL-60 cells. We further examined the protein–protein interactions by Zdock CO-IP and co-localization. As shown in Figure 2B, WT1 could bind to the CD34 protein in Zdock (Zdock prediction score = 1802.162). The protein–protein interactions between WT1 and CD34 were confirmed by CO-IP (Figure 2C) and co-localization (Figure 2D), indicating a correlation between WT1 and CD34 in maintaining the leukemic state.

### 2.2. Shikonin Is Screened as a WT1 Inhibitor

Shikonin had a wide anti-tumor effect on solid tumors [28] and also displayed anti-leukemic properties in AML primary cell lines [25], but the underlying mechanism has remained unclear. To find a WT1 inhibitor, molecular docking was used to predict the binding of shikonin to WT1. As shown in Figure 3A, shikonin had a binding ability to WT1 (affinity = −5.62 kcal/mol). This binding ability was verified by a drug affinity reaction target stability (DARTS) experiment in which the stability of the WT1 protein increased with the increase in shikonin concentrations (0.1–3.2 μM) due to the formation of shikonin–WT1 complexes (Figure 3B). In addition, we also found that shikonin could promote WT1 protein degradation compared to the control (0.1% DMSO) group (at 1 h, *p* < 0.01; at 2 h, *p* < 0.01; at 3 h, *p* < 0.01; at 4 h, *p* < 0.01) in the protein half-life experiment (Figure 3C). Further, shikonin was shown to down-regulate significantly the WT1 protein level in HL-60 cells in a dose-dependent manner, indicating a function of shikonin as a WT1 inhibitor.

### 2.3. Shikonin Promotes HL-60 Cell Differentiation

Because WT1 expression downregulation was considered as a differentiation-linked event, we observed the differentiation effect of shikonin on HL-60 cells. First, the effect of shikonin on leukemia cell viability was evaluated using the MTT assay. Shikonin treatment (0.00 μM, 0.16 μM, 0.31 μM, 0.63 μM, 1.25 μM, 2.50 μM, 5.00 μM, 10.00 μM, 20.00 μM, and 40.00 μM) for 48 h could significantly decrease HL-60 cell viability in a dose-dependent manner (Figure 4A) and its IC50 was 4.89 μM (4.13–5.64 μM). Based on these results, three nontoxic concentrations of shikonin (0.1 μM, 0.2 μM, and 0.4 μM) was selected as a WT1 inhibitor to treat Hl-60 cells. As shown in Figure 4B,C, shikonin at the experimental dose could significantly increase the cell size and volume in a dose-dependent manner in 48 h compared with the control group in Wright Giemsa staining and the Holomonitor system. Further, we found that shikonin could also down-regulate CD34 expression and increase CD11b expression in a dose-dependent manner in immunofluorescence (Figure 4D), suggesting an inducing effect of shikonin on HL-60 cells differentiating into mature granulocytes/monocytes. However, WT1 overexpression could reblock the differentiation effect of shikonin on HL-60 cells (Figure 4D), indicating a competition relation between WT1 and shikonin in HL-60 cell differentiation. This result was confirmed in Western blot (Figure 4E). Unlike shikonin, ATRA is a typical differentiation agent and also exerted a cytotoxic effect when it induced cell differentiation independent of WT1 in HL-60 cells (Figure 5A–C).

### 2.4. Shikonin-Differentiated HL-60 Cells Gradually Lose the Ability of Cell Proliferation and Self-Renewal

Theoretically, the differentiated and matured cells will finally lose the ability of cell proliferation and self-renewal ability. To explore the characteristics of shikonin-differentiated leukemia cells, HL-60 cells were treated with shikonin or ATRA for 10 d to detect the cell proliferation, cell cycle, apoptosis, mitochondrial membrane potential (MMP), and self-renewal. Compared with the control group, the EdU-positive cells were significantly reduced in the shikonin group in a dose-dependent manner (Figure 6A), and shikonin-differentiated HL-60 cells showed a cell cycle arrest at the G0/G1 phase (Figure 6B) and more cell apoptosis (Figure 6C). Simultaneously, shikonin treatment also significantly decreased cell MMP (Figure 6D) and soft agar clone formation (Figure 6E), indicating a progressive loss of cell proliferation and self-renewal ability in shikonin-differentiated HL-60 cells. ATRA-treated HL-60 cells had a similar fate (Figure 6A–E).

### 2.5. Shikonin Has Anti-Leukemia Activity In Vivo

To further confirm the anti-leukemia activity of shikonin, a leukemia mouse model was established in vivo. In this model, model mice showed increased WBC, NEU, and NEU% and decreased PLT and died one by one (Figure 7A–D). Shikonin could significantly decrease WBC, NEU, and NEU% and increase PLT in a dose-dependent manner compared with model mice (Figure 7A–C), and it also prevented mouse death (Figure 7D). Compared with the model group, shikonin could significantly decrease marrow megakaryocytes in bone marrow smears in Wright Giemsa staining (Figure 7E,G), and attenuated WT1- and CD34-positive cells in bone marrow cells in flow cytometry (Figure 7F,H). Arnebia euchroma (Royle) Johnst (AEJ) extract and ATRA (20 mg/kg) had a similar effect to 10 mg/kg of shikonin against leukemia; however, ATRA showed more toxicity indicated by the lower WBC, NEU, and marrow megakaryocytes compared to normal mice (Figure 7A–H).

## 3. Discussion

Leukemia is characterized by blocked differentiation and abnormal malignant proliferation [29]. Although most studies identified WT1 as an oncogene in AML, WT1 antisense oligonucleotide markedly reduced cell viability in myeloid MM6 cells and erythroid K562 cells but not in promyeloid HL-60 cells, although all of them had high WT1 expression [30], indicating a WT1 function dependent on the specific cell background. WT1 expression downregulation is considered as a differentiation-linked event in HL-60 cells [17]. In this paper, first, we confirmed WT1 protein expression in HL-60 cells. Next, WT1 was upregulated by Lentivirus transfection and its role was examined in HL-60 cells. It was shown that WT1 overexpression unaffected cell viability but could promote HL-60 cell resistance to H_2_O_2_. This result corresponds well with the opinion that WT1 overexpression only reflects a leukemic state although it is contradictory to the cycle modulating effect of WT1. WT1 can affect transcription by either its zinc-finger domain binding to DNA or its amino terminal interacting with other proteins; the balance between the DNA-binding effect and protein interaction determines WT’s fate as an activator or a repressor of other proteins in AML [31]. This could explain why WT1 downregulation did not affect cell viability, although it obviously increased cell differentiation markers.

CD34 is a cell surface sialomucin-like adhesion molecule and its expression is considered as a convenient marker of cancer stem cells (CSCs) and hematopoietic stem cells (HSCs), which maintain self-renewal and differentiation capacity [27]. It was reported that WT1 had a consistent up-regulation with CD34 in acute leukemia cells and also had a simultaneous down-regulation when leukemia cells were differentiated into mature cells [32]. Although their precise function remains unknown, we believe that there may be an interaction between WT1 and CD34. In this paper, we found that CD34 expression was simultaneously down-regulated in shikonin-treated HL-60 cells, indicating a role of WT1 in the immature phenotype in HL-60 cells. In Zdock, CO-IP, and co-localization, we confirmed the protein–protein interactions between WT1 and CD34 proteins, indicating a possible correlation between WT1 and CD34 in maintaining the leukemic state. A previous study showed that the cytoplasmic domain region of full-length CD34 was entirely responsible for stem properties in hematopoietic cells [33]. Scharnhorst’s study showed that the WT1 amino terminal domain is sufficient in modulating myeloid differentiation by interacting with p73 proteins [34]. In addition, it was verified that WT1 protein peptides could also be presented on the cell surface to become a cancer-associated antigen [35]. Therefore, we supposed that it was possible for the protein–protein interactions between WT1 and CD34 to be in charge of the immature cell phenotypes. However, this high hypothesis needs to be investigated by further studies.

It is well known that leukemia is a malignant clonal disease of HSCs or primitive progenitor cells; therefore, leukemia stem cells (LSCs) have become the key targets for eradicating leukemia [36,37,38]. Although LSCs are only a small population in AML, they are the root of initiation, malignant proliferation, and blocked differentiation in leukemia [39,40]. As with normal HSCs, LSCs also have multidirectional differentiation potential [41], and their cell surface antigens vary with cell differentiation and maturation [42]. According to the cell surface antigens, different cell types can be accurately identified [43]. It has been approved widely that CD11b is one of the most important surface antigens in mature granulocytes/monocytes [44]. In this study, we found that shikonin could significantly promote CD11b expression when it suppressed WT1 and CD34 expression, suggesting a preventing effect of WT1 and CD34 on granulocyte/monocyte maturation in HL-60 cells. In addition, in this study, the untreated HL-60 cells also expressed CD11b to some extent, indicating its differentiation potential into granulocytes/monocytes; this may explain why differentiation therapy easily succeeds in AML cells.

Shikonin presents antitumor activity against various type of cancers including hematological malignancies [45]. To explore whether shikonin can inhibit WT1 to restore leukemia cell differentiation, we use molecular docking and DARTS to verify the binding of shikonin to the WT1 protein. Further, the suppressing effect of shikonin on the WT1 protein was proved by a protein half-life experiment and Western blot. As expected, nontoxic shikonin as a WT1 inhibitor could decrease the CD34 protein level and simultaneously promoted CD11b protein expression in normal HL-60 cells in 48 h, whereas WT1 overexpression could reblock the differentiation effect of shikonin on HL-60 cells, indicating a competition relation between WT1 and shikonin in HL-60 cell differentiation. It is reported that the ultimate fate of differentiated cells is natural apoptosis [46]. In order to further observe whether shikonin-differentiated HL-60 cells finally lose the ability of cell proliferation and self-renewal, the cell culture was prolonged for 10 d after shikonin addition. We found that shikonin-differentiated HL-60 cells showed low proliferation and MMP, cell cycle arrest, weak self-renewal, and high apoptosis. These results proved our hypothesis and also indicated a leukemia-preventing effect of shikonin on WT1-expressed leukemia cells. The preventing activity of shikonin on leukemia was further confirmed in a leukemia mouse model, in which shikonin could improve WBC and PLT and reduced mortality. Unlike ATRA suppressing WBC and marrow cells, compared to normal mice, shikonin only decreased WBC and marrow cells compared to leukemia mice, implying a high selectivity.

This study is the initial WT1-oriented drug design. Our object is to find a WT1 inhibitor to restore leukemia cell differentiation. Although we only used the faster but less reliable techniques (molecular docking, DARTS experiment, and protein half-life experiments) to screen shikonin as a WT1 inhibitor, our attention focused on its biological efficacy but not structure analysis. As for the shikonin–WT1 interaction, limited by the lack of sophisticated equipment and well-trained specialists, we did not use more precise methods such as isothermal titration calorimetry (ITC, a method for measuring the binding event’s thermodynamic parameters) and microscale thermophoresis (MST, a method for measuring the binding event’s dissociation constants) to characterize the shikonin-WT1 interaction [47,48]. In the future, we will cooperate with biophysicists to address protein–ligand interactions by those well-established biophysical methods.

The bleeding and fatigue symptoms often occur in AML patients due to therapeutic toxicity to normal tissues [49]. According to TCM theory, Zicao can cool the blood and relieve internal heat, whereas shikonin is an active component representing Zicao functions [50]. In this study, shikonin at the selected concentrations could suppress the WT1 protein to restore HL-60 cell differentiation without killing cells, implying a low toxicity to normal tissues. Although further investigations in a series of AML cells are required, our results suggested that shikonin could be a lead compound as WT1 inhibitor. The interaction between shikonin and WT1 will help design more effective and specific WT1 inhibitors by molecular dynamics calculation and structure drug design. Preclinical studies and preliminary clinical trials provide the possibility for shikonin-type drugs as anti-cancer drugs [51,52], and it is expected that shikonin itself and its natural or synthetic derivatives can be considered as the novel differentiation agents to fight AML being unresponsive to current chemotherapeutic drugs.

## 4. Materials and Methods

### 4.1. Reagents

AEJ was purchased from Zhangzhongjing Pharmacy (Zhengzhou, China) and its name was reviewed using the Plant List website (http://www.theplantlist.org, accessed on 29 April 2021). Shikonin (purity > 98%, verified by HPLC) was purchased from Manster Biotechnology Co., Ltd. (A0186, Chengdu, China) and its chemical structure is shown in Figure 3A. RPMI 1640 medium (R0883), ATRA (PHR1187), and cycloheximide (CHX, C7698) were purchased from Sigma Biotechnology Co., Ltd. (Saint Louis, MO, USA). Fetal bovine serum (FBS) was bought from Zhejiang Tianhang Biotechnology Co., Ltd. (11011-8611, Zhejiang, China). 3-(4, 5-dimethylthiazol-2-yl)-2, 5-diphenyltetrazolium bromide (MTT) was bought from Saiguo Biotechnology Co., Ltd. (298-93-1, Guangzhou, China). The Wright Giemsa staining kit was purchased from Beijing Reagan Biotechnology Co., Ltd. (DM0007, Beijing, China). Primary antibodies WT1 (sc-7385) and β-actin (66009-1-Ig) were purchased from Santa Cruz Biotechnology Co., Ltd. (Santa Cruz, CA, USA). APC-labeled anti-CD34 (14486-1-AP) and FITC-labeled anti-CD11b (sc-1186) were purchased from Proteintech Group, Inc. (Wuhan, China). The 5-ethynyl-2′-deox-yuridine (EdU) kit (C10310-3) was purchased from RiboBio Co., Ltd. (Guangzhou, China). The RT-PCR kit was purchased from Accurate Technology Co., Ltd. (AG11701, Hunan, China). M-PER lysis reagents (78501) and the Co-IP Kit (88804) were purchased from Thermo Fisher Scientific Biotechnology Co., Ltd. (Waltham, MA, USA). The Cell Cycle Assay Kit (PI/RNase Staining) was purchased from Dojindo Chemical Technology Co., Ltd. (C543, Shanghai, China). Streptomycin/penicillin (P1400), the cell apoptosis assay kit (Annexin V-FITC/PI, CA1020), and agarose (A8350 and A8201) were purchased from Solarbio science and technology Co., Ltd. (Beijing, China). The JC-1 staining kit (C2005) and X-Gal staining (ST912) were purchased from Beyotime Biotechnology Co., Ltd. (Beijing, China). Anti-Mouse (CTS002) and Anti-Rabbit HRP-DAB cell staining kits (CTS005) were purchased from R&D Systems Inc. (Minneapolis, MN, USA). All other chemicals and solvents were of analytical grade or better. AEJ extract was produced by a previous method [23].

### 4.2. Cell Culture

HL-60 cells, an acute myeloid leukemia cell line, were purchased from ATCC (Manassas, VA, USA). The cells were cultured in RPMI 1640 medium (containing 10% FBS, 100 μg/mL of streptomycin, and 100 U/mL of penicillin). WT1 normally expressed or highly expressed HL-60 cells were in a humidified atmosphere with 5% CO_2_ at 37 ℃.

### 4.3. Lentivirus Transfection

WT1 overexpression lentivirus plasmid was prepared according to the GenePharma instruction. HL-60 cells at 4 × 10^3^ cells/mL were transfected by virus and Polybrene (5 μg/mL) in 96-well plates. Fluorescence expression was used to test the transfected efficiency in 72–96 h after infection. The transfected cells were screened by 0.7 µg/mL of puromycin, cloned via limited dilution and amplified as HL-60O cells. Western blot was used to test WT1 expression.

### 4.4. Western Blot

Western blot was performed as in the previous study [53]. In short, a RIPA lysis buffer containing mixed protease and phosphatase inhibitors (100:1) was used for total protein extraction, and the protein concentration was determined by the BCA method. The extracted protein was used to prepare the sample for SDS gel electrophoresis. The target protein and the loading control were run on the same gel. Finally, the optical density of the band on the film was quantified by one 4.0.26 software.

### 4.5. DNA Injury

Cells were treated with 100 μM H_2_O_2_ for 24 h based on a trial test in which cells exhibited the obvious apoptosis and DNA injury. For DNA injury assay, 20 μL of cells (5 × 10^3^) were embedded in 80 μL of 0.75% low-melting point agarose on slides pre-coated with 100 μL of 1% normal melting point agarose. After incubated in lysis solution for 1 h at 4 °C in the dark, the slides were placed on an electrophoresis box containing an alkaline solution and exposed to the alkali for 40 min for DNA unwinding. Electrophoresis was performed for 15 min and the slides were stained with 20 μg/μL of PI. The comets were visualized under an Olympus fluorescence microscope, and Comet 5.0 was used to analyze the percentage of tail DNA in total DNA in every 100 cells. For the cell apoptosis assay, cells were collected, stained by the Annexin V-FITC/PI dual staining kit, and measured using flow cytometry.

### 4.6. The Assays of Cell Viability, Morphology, and Differentiation Markers

MTT: HL-60 cells were collected in the logarithmic growth stage, and the cell density was adjusted to 10^6^ cells/mL and inoculated in 96-well plates for 0, 6, 12, 18, 24, 30, 36, 42, and 48 h to calculate the doubling time by the least squares fit using graphpad prism 5.0. The cells were treated with different concentrations of shikonin or ATRA for 48 h. After 20 μL of MTT was added, cells continued to incubate for 4 h. An amount of 150 μL of dimethyl sulfoxide was added, the absorbance was detected by a full-wavelength reader, and the cell viability in the treated group was compared to the untreated group.

Wright Giemsa staining: HL-60 cells were treated with 0.1 μM, 0.2 μM, and 0.4 μM shikonin for 48 h. After incubation, cells were collected for preparation of cell smears. For bone marrow cells, after 43 days of treatment, the mice were killed, the bone marrow was dissected, and the bone marrow smear was prepared through a 70-mesh sieve. Finally, the smears were cooled, dried, and fixed in methanol for 5 min. Wright Giemsa staining was performed, and picture information was observed under a microscope (Olympus, Japan).

HoloMonitor system: The Holomonitor system was used to monitor the cell morphological changes. HL-60 cells were cultured as in the Wright Giemsa staining assay. After incubation, the cell size was observed and recorded in a 3D level by the M4 laser holographic cell imaging and analysis system Holomonitor system (Holomonitor, Phiab, Sweden). Holostudio™ 2.3 software (Elastic Compute Service, Beijing, China) was used for data analysis.

Immunofluorescence: HL-60 cells treated with shikonin or ATRA for 48 h were collected for cell smear. The smears were immersed in 4% paraformaldehyde for 15 min, infiltrated with 0.1% Triton X-100 for 10 min, and blocked with 5% bovine serum albumin (BSA) for 2 h. Then, the cell smears were incubated with APC-labeled anti-CD34 (1:100) and FITC-labeled anti-CD11B (1:100) at 37 °C in the dark for 30 min. The fluorescence signal was detected by a Leica fluorescence microscope. The relative fluorescence intensity of cell target proteins was analyzed by Image J software (OLYMPUS, Japan).

### 4.7. The Interactions of CD34 Protein–WT1 Protein and Shikonin–WT1 Protein

Zdock predicts: The protein–protein interaction between CD34 and WT1 was predicted by Zdock software (version 3.0.3, http://zdock.umass34med.edu/, accessed on 12 March 2021) [54]. The binding ability of protein to protein was evaluated by the prediction score.

Molecular docking: The interaction between shikonin and the WT1 protein was predicted by molecular docking. The mol2 molecular structure file of shikonin was obtained from the PubChem database (https://pubchem.ncbi.nlm.nih.gov/, accessed on 12 March 2021). The 3D structure file of the WT1 protein was downloaded from the RCSB PDB database (https://www.rcsb.org, accessed on 12 March 2021). The ligand within the crystal structure complex was extracted by PyMOL software. Then, Autodock 4.2 software was used for molecular docking according to the structure files of small-molecule drugs and proteins. The results with high docking score and stable conformation were transformed into PDB format, and the visual analysis of molecular docking was carried out using PyMOL 2.3.2 software.

### 4.8. Co-Immunoprecipitation

HL-60 cells were collected and rinsed twice with PBS. M-PER protein lysis solution was added to the sample tube and incubated on ice for 1 h. The total protein was incubated with anti-WT1 antibody or IgG (as negative control) overnight at 4 °C to form an antigen sample–antibody mixture. A/G magnetic beads were added into an EP tube, and the magnetic beads were rinsed twice with immunoprecipitation rinsing buffer. The antigen sample–antibody mixture was incubated with magnetic beads. Immunoprecipitation rinsing buffer was added, the magnetic beads were rinsed twice, and supernatant was discarded. The magnetic beads were rinsed once with ultrapure water. Loading buffer was added into the magnetic beads to prepare samples for Western blot analysis. For co-localization, HL-60 cells were incubated with APC-labeled anti-WT1 (1:100) and FITC-labeled anti-CD34 (1:100) at 37 °C in the dark for 30 min, and confocal microscopy (Nikon) was used to observe co-localization between CD34 and WT1.

### 4.9. Drug Affinity Responsive Target Stability (DARTS)

The principle of DARTS technology is that the stability of proteins will be enhanced if small molecules can combine with the target protein [55]. A quantity of 1 × 10^7^ HL-60 cells were placed in a 1.5 mL EP tube. M-PER protein lysis solution was added to each sample tube and incubated on ice for 1 h. The cell lysis was centrifuged at 14,000 rpm for 10 min and the protein was quantified by the BCA quantitative method. Total protein and different concentrations of shikonin (0 mΜ, 0.1 μM, 0.2 μM, 0.4 μM, 0.8 μM, 1.6 μM, and 3.2 μM) were mixed and incubated at 4 ℃ for 1 h. After incubation, streptomyces protease was added to each sample tube and incubated at 25 °C for 15 min. Finally, 1× loading buffer was added into each sample tube and the sample tube was boiled at 100 °C for 5 min to terminate the reaction. The samples were analyzed by Western blot.

### 4.10. Protein Half-Life Assay

HL-60 cells were seeded at a density of 1 × 10^6^/ mL in 25 cm^2^ cell culture flasks and treated with 10 μM cycloheximide (CHX) plus 0.2 μM shikonin or 0.1% DMSO for 0, 1, 2, 3, and 4 h. Finally, the proteins were analyzed by Western blot.

### 4.11. Cell Proliferation and Mitochondrial Membrane Potential (MMP) Assays

HL-60 cells were treated with shikonin (0.1 μM, 0.2 μM, and 0.4 μM) or ATRA (5 μM) for 10 d to maintain the differentiation state, and the medium was changed every other day. Cell proliferation and MMP were assessed, respectively, using the Cell-Light EdU Apollo488 Kit and JC-1 staining kit following manufacturer’s instructions. Images were captured using an Olympus fluorescence microscope (Olympus, Japan), and the percentage of Edu-positive cells or JC-1 staining intensity was calculated from 6 random fields using Image J software.

### 4.12. Cell Cycle and Apoptosis Assays

Shikonin or ATRA-treated HL-60 cells were cultured as in cell proliferation and MMP assays. After cells were cultured, the cell cycle and apoptosis were assessed by a cell cycle assay kit and Annexin V-FITC/PI dual staining kit using flow cytometry following manufacturer’s instructions.

### 4.13. Cell Self-Renewal Assay

Shikonin or ATRA-treated HL-60 cells were suspended in 0.5 mL of RPMI 1640 Medium containing 20% FBS and 0.5 mL of top agar (0.7%) and transported to 6-well plates precoated with 0.5 mL of bottom agar (1.2%). After being inoculated for 10 d, cells were stained with crystal violet, images were captured using an Olympus microscope (Olympus, Japan), and six random regions were selected in every well to count the number of colonies.

### 4.14. Mouse Leukemia Model

A mouse leukemia model was developed according to the method of Ma et al. [56]. Six-week-old female NOD/SCID mice, weighing 22–24 g, were obtained from Charles River Laboratory Animal Technology Co., Ltd. (Beijing, China; license: SCXK (jing) 2016-0011). The mice were irradiated by a medical linear accelerator with 2.4 Gy. At 24 h after mice were irradiated, HL-60 cells at a density of 2.5 × 10^7^/mL were injected into the tail vein (0.2 mL/mouse, *n* = 20/group). The next day, mice received shikonin (20, 10, and 5 mg/kg/day) or AEJ extract (equal to AEJ 2.0 g/kg/day or shikonin 10 mg/kg/day) via intragastric administration once a day, or ATRA (20 mg/kg/day) via intraperitoneal injection once every two days until the experiment ended (7 weeks). All mice were kept in a standard, specific-pathogen-free (SPF) facility of the animal laboratory with 12 h dark/light cycle, controlled temperature (24 °C–25 °C), and relative humidity (40–70%) conditions. All animal treatments were approved by the Henan University Animal Care and Use Committee (Permission number: HUSAM 2021-069, 20210309), and conducted in accordance with the ethical standards and national guidelines. Mice were sacrificed when half of the mice in the model group died, routine blood analysis was carried out by an automatic blood cell analyzer (Mindray, Nanjing, China), and the bone marrow cells were extracted for cell smears and flow cytometry analysis. The survival curves were recorded.

### 4.15. Data Statistical Analysis

The data were presented as mean ± standard deviation (SD) and statistically analyzed using GraphPad Prism 9.0 (San Diego, CA, USA). Groups were compared using a two-tailed unpaired Student’s *t*-test or one-way analysis of variance (ANOVA) with the Bonferroni Multiple Comparison Test or Dunnett’s Multiple Comparison Test. Differences with a *p* value less than 0.05 were considered statistically significant.

## 5. Conclusions

This study proved the suppressing effect of shikonin on WT1 in protein expression and function and suggested that: (1) shikonin can down-regulate the WT1 protein level for leukemia differentiation therapy, and (2) the interaction between WT1 and CD34 proteins may be responsible for granulocyte/monocyte immaturity in HL-60 cells.

## Figures and Tables

**Figure 1 molecules-27-08264-f001:**
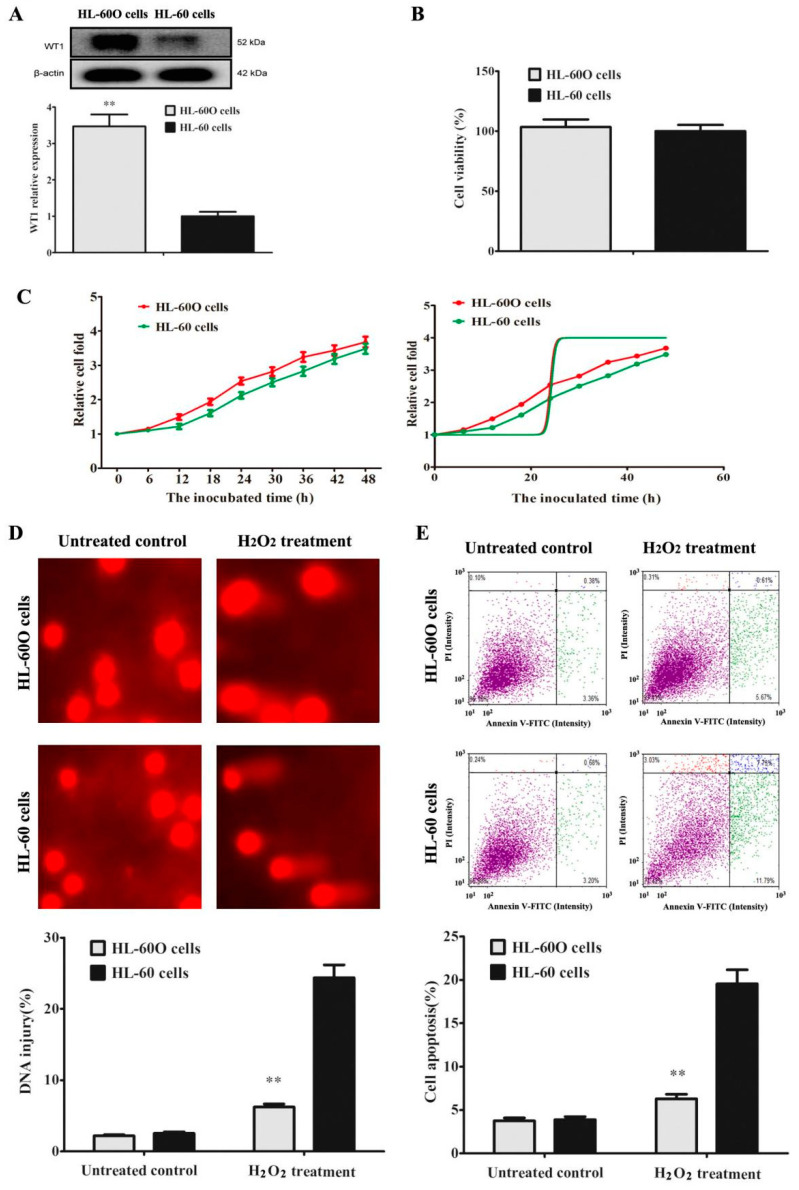
WT1 over-expression promotes HL-60 cell resistance to H_2_O_2_. (**A**) WT1-overexpressed HL-60 cells (HL-60O cells) were established by Lentivirus transfection, and WT1 expression was detected by Western blot (*n* = 3). The equivalent cells were incubated for 48 h. (**B**) Cell viability was detected by MTT assay (*n* = 6). (**C**) Relative cell fold was detected by MTT assay. Cells were treated with 100 μM H_2_O_2_ for 24 h. (**D**) DNA injury was indicated by the percentage of tail DNA in total DNA in Comet assay (*n* = 3). (**E**) Cell apoptosis was tested by Annexin V-FITC/PI dual staining in flow cytometry (*n* = 3). The data were presented as the mean ± SD. ** *p* < 0.01 compared to the parent HL-60 cells.

**Figure 2 molecules-27-08264-f002:**
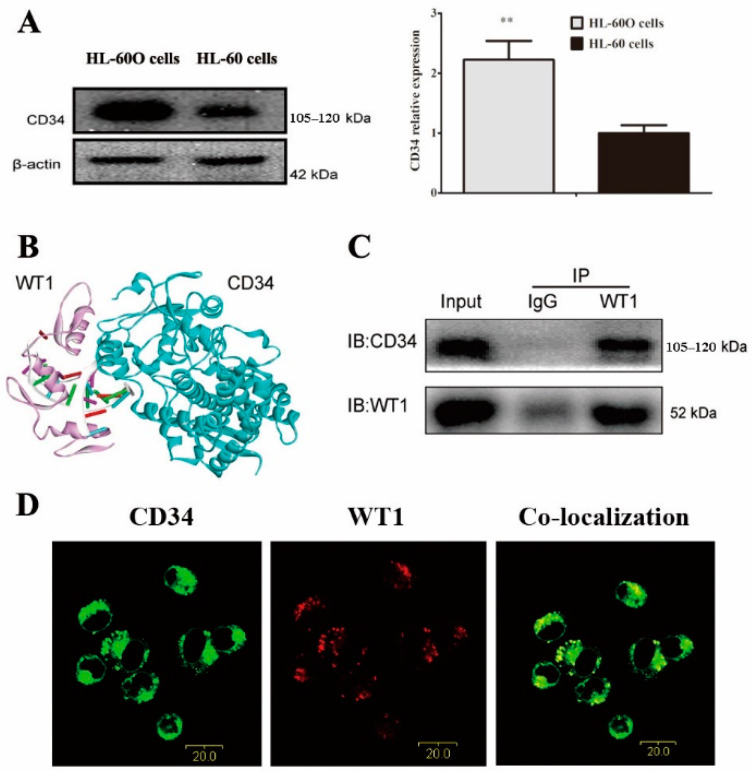
The protein–protein interactions between WT1 and CD34 protein. (**A**) CD34 protein expression was detected by Western blot in HL-60 cells and HL-60O cells (*n* = 3). The data were presented as the mean ± SD (*n* = 3). ** *p* < 0.01 compared to the parent HL-60 cells. The protein–protein interactions between WT1 and CD34 were indicated by (**B**) Zdock (the red lines represent Zn finger) and (**C**) Western blot in CO-IP experiment, and (**D**) co-localization in immunofluorescent staining.

**Figure 3 molecules-27-08264-f003:**
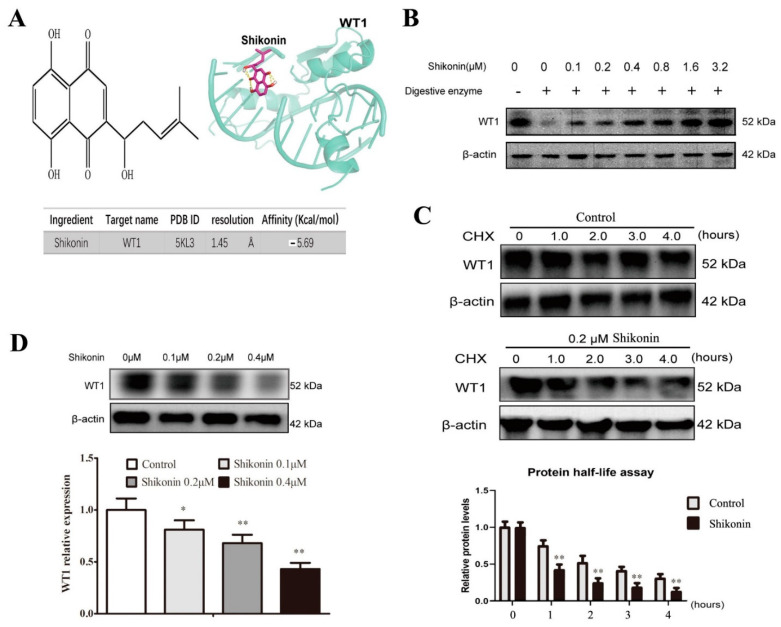
Shikonin is screened as a WT1 inhibitor. (**A**) The chemical structure of shikonin and its binding affinities to WT1 in molecular docking. (**B**) The binding of shikonin to WT1 in DARTS experiment. (**C**) The effect of shikonin on WT1 protein degradation was detected by the protein half-life test. (**D**) Shikonin suppressed WT1 protein expression indicated by Western blot in HL-60 cells (*n* = 3). Cells were incubated with shikonin for 48 h. The data were presented as the mean ± SD. * *p* < 0.05; ** *p* < 0.01 compared to the untreated control.

**Figure 4 molecules-27-08264-f004:**
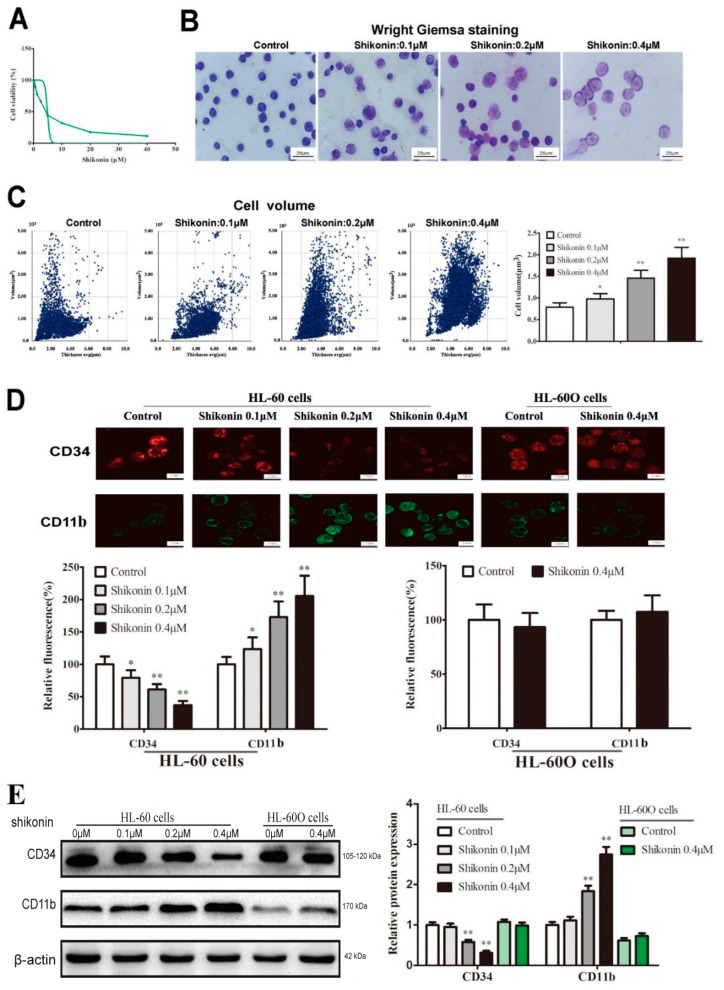
Shikonin promotes leukemia cell differentiation. HL-60 cells were treated with shikonin for 48 h. (**A**) Cell viability was detected by MTT assay. (**B**) Cell morphology was detected by Wright Giemsa staining. (**C**) The cell volume distribution was detected by the Holomonitor system. (**D**) The expression of CD34 and CD11b was detected by immunofluorescence in HL-60 cells. (**E**) CD34 and CD11b protein expression was detected by Western blot. The data were presented as the mean ± SD (*n* = 3). * *p* < 0.05; ** *p* < 0.01 compared to the untreated control.

**Figure 5 molecules-27-08264-f005:**
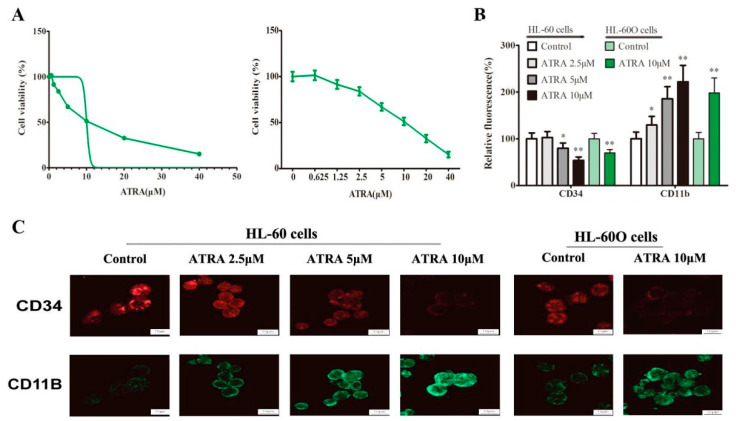
ATAR-induced HL-60 cell differentiation is associated with cytotoxicity. (**A**) ATAR decreases cell viability in a dose-dependent manner. (**B**,**C**) The expression of CD34 and CD11b was detected by immunofluorescence in HL-60 cells and in HL-60O cells. The data were presented as the mean ± SD (*n* = 3). * *p* < 0.05; ** *p* < 0.01 compared to the untreated control.

**Figure 6 molecules-27-08264-f006:**
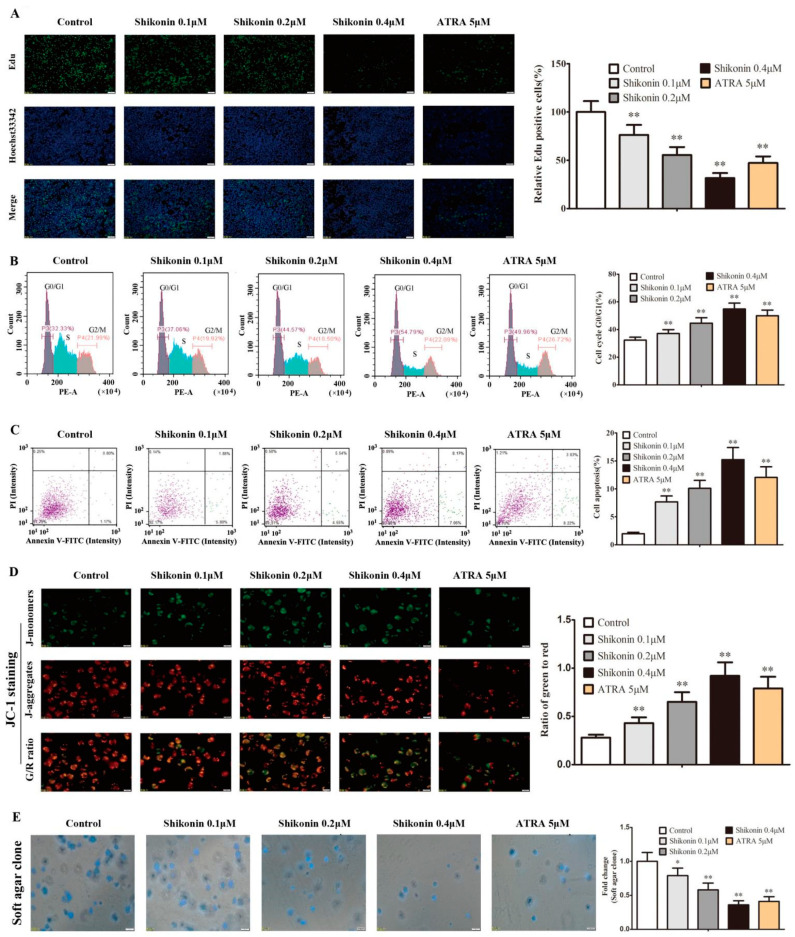
Shikonin-differentiated HL-60 cells gradually lose the ability of cell proliferation and self-renewal. HL-60 cells were treated with shikonin for 10 d. (**A**) Cell proliferation was detected by EdU incorporation. (**B**) The cell cycle was detected by PI staining using flow cytometry. (**C**) Cell apoptosis was detected by Annexin V-FITC/PI staining using flow cytometry. (**D**) The mitochondrial membrane potential (MMP) was detected by JC-1 staining, which forms J-monomers with green fluorescence to label the low MMP and J-aggregates with red fluorescence to label the high MMP. (**E**) Cell self-renewal was detected by soft agar clone formation. The data were presented as the mean ± SD (*n* = 3). * *p* < 0.05; ** *p* < 0.01 compared to the untreated control.

**Figure 7 molecules-27-08264-f007:**
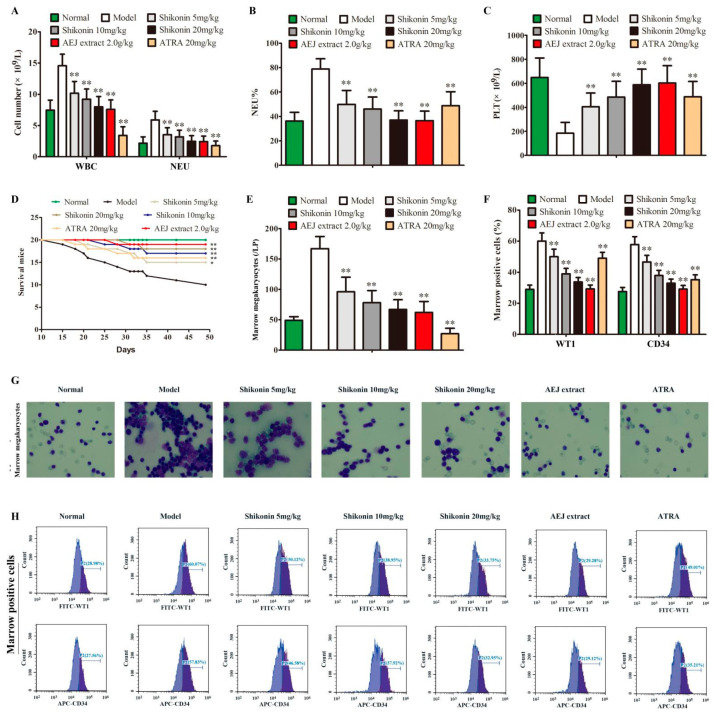
Shikonin has anti-leukemia activity in vivo. Leukemia mice were treated for 7 weeks, and the blood routine was examined. (**A**) White blood cells (WBCs) and neutrophils (NEUTs) (*n* = 10). (**B**) Percentage of neutrophils (NEU%) (*n* = 10). (**C**) Platelet (PLT) detected. (**D**) Mouse survival curve (*n* = 20). (**E**,**G**) Marrow megakaryocytes in marrow smears indicated by wright Giemsa staining (*n* = 5). (**F**,**H**) WT1- and CD34-positive cells in marrow cells detected by flow cytometry (*n* = 5). The data were presented as mean ± SD. * *p* < 0.05; ** *p* < 0.01 compared to the untreated model.

## Data Availability

The data that support the findings of this study are available from the corresponding author (10200029@vip.henu.edu.cn) upon reasonable request.

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
