# Peer review of "Shikonin as a WT1 Inhibitor Promotes Promyeloid Leukemia Cell Differentiation"

_molecules, 2022, doi:10.3390/molecules27238264_

Round 1

Reviewer 1 Report

The authors presented an interseting study about shikonin and WT-1 in AML cell differentiation. However they should improve the manuscript as detailed below:

Structure of shikonin should be, instead in Fig 1a, placed in Fig 3 (maybe it should be 3a)

ATAR in Fig 5 legend should be corrected. Also, I cannot see the point of Fig 5? If Fig 5 remains maybe you need information how ATRA affect differentiation of HL-60O cells?

 CD11B should be replaced with CD11b in text and figures.

Do you mean bone marrow megakaryocytes instead karyocytes?

MMP abbreviation should be introduced in a row 160

Markers of rows in immunofluorescence in Fig6D should be added

The colors in graph bars in Fig 7 do not correspond to colors in figure legends

Include manufacturer for JC-1 staining kit, cell cycle assay kit and Annexin V-FITC/PI dual staining kit

Author Response

The authors presented an interseting study about shikonin and WT-1 in AML cell differentiation. However they should improve the manuscript as detailed below:

  1. Structure of shikonin should be, instead in Fig 1a, placed in Fig 3 (maybe it should be 3a)

Response: This is a very useful comment. Based on this comment, we instead structure of shikonin in Fig 1a and placed it in Fig 3a in the revised manuscript.

   Thanks again for this good advice.

  1. ATAR in Fig 5 legend should be corrected. Also, I cannot see the point of Fig 5? If Fig 5 remains maybe you need information how ATRA affect differentiation of HL-60O cells?

Response: This is a very useful comment. Based on this comment, we corrected Fig 5 legend and supplied information how ATRA affect differentiation of HL-60O cells   in the revised Fig 5.

   Thanks again for this good advice.

  1. CD11B should be replaced with CD11b in text and figures.

 Response: This is a very useful comment. Based on this comment, we replaced all CD11B with CD11b in text and figures in the revised manuscript.

   Thanks again for this good advice.

  1. Do you mean bone marrow megakaryocytes instead karyocytes?

Response: This is a very useful comment. Based on this comment, we instead karyocytes with megakaryocytes in Fig 7 and in the revised manuscript.

   Thanks again for this good advice.

  1. MMP abbreviation should be introduced in a row 160

Response: This is a very useful comment. Based on this comment, we added MMP abbreviation in the first appearance in the revised manuscript .

   Thanks again for this good advice.

  1. Markers of rows in immunofluorescence in Fig6D should be added

Response: This is a very useful comment. Based on this comment, we added Markers  in revised Fig6D and statement in the revised figure 6 legend.

   Thanks again for this good advice.

  1. The colors in graph bars in Fig 7 do not correspond to colors in figure legends

Response: This is a very useful comment. Based on this comment, we adjusted the colors in graph bars in Fig 7 to correspond to colors in figure legends .

   Thanks again for this good advice.

  1. Include manufacturer for JC-1 staining kit, cell cycle assay kit and Annexin V-FITC/PI dual staining kit

Response: This is a very useful comment. Based on this comment, we supplied manufacturer for JC-1 staining kit, cell cycle assay kit and Annexin V-FITC/PI dual staining kit in Reagents in line 335-340 in the revised manuscript.

   Thanks again for this good advice.

Reviewer 2 Report

The manuscript submitted by Guo and colleagues aimed to investigate the implication of Wilm’s tumor 1 (WT1) in the molecular mechanism via which shikonin promotes promyeloid leukemia cell differentiation. They showed that (1) overexpression of WT1 does not affect cell viability but promoted resistance to H2O2; (2) shikonin reduced the expression and half-life of WT1; (3) it promoted leukemia cell differentiation and the loss of the ability of cell proliferation and self-renew; (4) and it has anti-leukemia activity in vivo, too.

Apoptosis- and differentiation-inducing effects of shikonin in HL-60 cells are not entirely novel (Ref. [18] and DOI:10.1155/2012/781516), while in vivo anti-leukemic effects of shikonin were not demonstrated in the literature before. The effect of shikonin on WT1 protein level was plausibly demonstrated. However, there is not a strong evidence of the correlation of WT1 and CD34 expression and that of their protein-protein interaction (see specific comments below).

The English of the manuscript (especially in the Introduction part) is confusing and difficult to understand at several points. It contains numerous grammatical and stylistic errors. The Introduction should provide more background information to understand the goals of the study. For example, what are the targets of WT1? What gives the idea that shikonin may regulate WT1 protein level?

The quality of certain microscopic and Western blot images are not satisfactory (see comments below).

Specific comments:

Figure 1

(1) Fig. 1A: The chemical structure of shikonin does not belong to this figure. It should be shown in figure 3.

(2) Fig. 1D: How cell proliferation was determined for Fig. 1D? Did you apply Cell-Light EdU Apollo488? It should be specified in the figure legend. What is the unit for the y axis? Is it relative proliferation? Error bars are missing from the graph.

(3) Fig. 1E: Image of the comet assay for H2O2-treated HL-60 cells is not typical. Based on the theory of comet assay, comets are expected to be aligned along the same axis, while here they have different orientations. More convincing microscopic images would be required to prove DNA injury. How was the percentage of DNA injury calculated?

(4) According to Materials and methods, 100 μM H2O2 for 24h was applied before comet assay. Isn’t it extreme? Did you use the same conditions for apoptosis assay (Fig. 1F)?

(5) What would be the effect of shikonin on H2O2-induced DNA injury in HL-60 cells?  As an antioxidant, shikonin is expected to attenuate H2O2-induced oxidative injury (Zhong et al DOI: 10.3892/etm.2021.10552). WT1 overexpression is proposed to protect the cells from oxidative stress (Fig. 1E). At the same time, shikonin is supposed to lower the level of WT1 (Fig. 3). How are these two effects compatible with one another?

Figure 2

These experiments aimed to test correlation between WT1 and CD34 expression (Fig. 2A) and protein-protein interaction of WT1-CD34 (Fig. 2C).

(6) Fig. 1A: The representative blot does not show significant change in CD34 expression, while a 50% increase is indicated by the relative expression graph. Further independent experiments would be required to strengthen this data.

(7) Fig. 2B: What is represented by the red lines in the molecular model?

(8) Fig. 2C: The only experimental proof of WT1-CD34 interaction is this co-IP experiment. The weakness of this experiment it that the IgG control IP sample also reacts with the anti-WT1 antibody. How did you eliminate detection-interference from the heavy-chain (approx. 50kDa) IgG-fragments of the antibody used for the initial immunoprecipitation? Full blots should be shown here to avoid potential misunderstandings. Alternative methods to verify co-localization and interaction of these to proteins would be required (eg. immunofluorescent staining, proximity ligation assay).

Figure 3

(9) Fig. 1B: Loading control is of bad quality.

(10) Fig. 1D: How long did you incubate the cells with shikonin before studying the expression of WT1?

(11) What is the supposed mechanism by which shikonin affects both the expression and the protein degradation of WT1?

 Figure 4

(12) Fig. 4D: Immunofluorescent images are hardly visible. Better-quality images would be needed to support the quantitative data. Alternatively, mRNA expression or Western blot study of CD34 and CD11b could be applied to verify differentiation.

(13) line 139: ‘the best differentiation effect of 0.4μM shikonin on HL-60 cells was able to be aborted in WT1 overexpressed HL-60O cells’. Could higher doses of shikonin induce differentiation in WT1-overexpressing cells? What concentration of shikonin would be needed to downregulate WT1 in overexpressing cells? These answers would strengthen the idea that shikonin induces HL-60 cell differentiation in a WT1-dependent menn

Figure 6

(14) Fig. 6D: What does the green and red color represent during JC-1 staining? Labels should be given next to the images.

Figure 7

(15) Fig. 7A-F: The color code of Fig. 7D is applied for all graph labelling, but for the other graphs the ‘model’ is labelled with white (instead of black) and ‘shikonin 10 mg/kg’ is labelled with black (instead of blue).

(16) line 222: ‘It was reported that WT1 had a consistent expression or lost with CD34 as cells reverse or differentiate [27].’ This is a confusing sentence. Please, clarify.

(17) line 359: The method is described in imperative mode (eg. add, collect, etc), which is unusual in research articles.  Please, modify.

(18) line 399 and 411: M-PER protein lysis buffer (not lysate)  

(19) line 413: “Total protein and different concentrations of shikonin … were mixed and incubated at 4° C for 10 h.” Why there is a need for that long incubation?

(20) I have identified some irrelevant references. For example, for WT1 antisense oligonucleotide, reference [23] should be cited. Ref. [40] is about Isothermal Titration Calorimetry, which is absolutely irrelevant to the sentence where it is mentioned (line 259). Isn’t there a shift in reference list? A thorough review of would be necessary.

In summary, I suggest that this paper could be accepted for publication in Molecules only after major revision (extensive editing of English; providing better quality microscopic images; providing full blot of co-IP experiment; alternative method to prove CD34-WT1 interaction; corrections).

Author Response

The manuscript submitted by Guo and colleagues aimed to investigate the implication of Wilm’s tumor 1 (WT1) in the molecular mechanism via which shikonin promotes promyeloid leukemia cell differentiation. They showed that (1) overexpression of WT1 does not affect cell viability but promoted resistance to H2O2; (2) shikonin reduced the expression and half-life of WT1; (3) it promoted leukemia cell differentiation and the loss of the ability of cell proliferation and self-renew; (4) and it has anti-leukemia activity in vivo, too.

Apoptosis- and differentiation-inducing effects of shikonin in HL-60 cells are not entirely novel (Ref. [18] and DOI:10.1155/2012/781516), while in vivo anti-leukemic effects of shikonin were not demonstrated in the literature before. The effect of shikonin on WT1 protein level was plausibly demonstrated. However, there is not a strong evidence of the correlation of WT1 and CD34 expression and that of their protein-protein interaction (see specific comments below).

The English of the manuscript (especially in the Introduction part) is confusing and difficult to understand at several points. It contains numerous grammatical and stylistic errors. The Introduction should provide more background information to understand the goals of the study. For example, what are the targets of WT1? What gives the idea that shikonin may regulate WT1 protein level?

Response: This is a very useful comment. Based on this comment, we do our best to correct numerous grammatical and stylistic errors and revise introduction for provide more background information to understand the goals of the study in line 55-62 and 64-79 in the revised manuscript.

   Thanks again for this good advice.

The quality of certain microscopic and Western blot images are not satisfactory (see comments below).

Specific comments:

Figure 1

  • 1A: The chemical structure of shikonin does not belong to this figure. It should be shown in figure 3.

Response: This is a very useful comment. Based on this comment, we instead structure of shikonin in Fig 1a and placed it in Fig 3a in the revised manuscript.

   Thanks again for this good advice.

  • 1D: How cell proliferation was determined for Fig. 1D? Did you apply Cell-Light EdU Apollo488? It should be specified in the figure legend. What is the unit for the y axis? Is it relative proliferation? Error bars are missing from the graph.

Response: This is a very useful comment. Based on this comment, we supply how cell proliferation was determined for Fig. 1D , specified it in the figure legend, and also supplied Error bars in the revised Fig.1.

   Thanks again for this good advice.

  • 1E: Image of the comet assay for H2O2-treatedHL-60 cells is not typical. Based on the theory of comet assay, comets are expected to be aligned along the same axis, while here they have different orientations. More convincing microscopic images would be required to prove DNA injury. How was the percentage of DNA injury calculated?

Response: This is a very useful comment. Based on this comment, we supplied the typical image of the comet assay in revised Fig 1. and also stated how  the percentage of DNA injury was calculated in methods in line 370-372 in the revised manuscript and figure 1 legend.

   Thanks again for this good advice.

  • According to Materials and methods, 100 μM H2O2for 24h was applied before comet assay. Isn’t it extreme? Did you use the same conditions for apoptosis assay (Fig. 1F)?

Response: This is a very useful comment. Why 100 μM H2O2 for 24h was applied was based on a trial test in which cells appeared the obvious apoptosis and DNA injury. We used the same conditions for comet assay and apoptosis assay. We state this condition in the revised manuscript line in 364-365 and in figure 1 legend.

   Thanks again for this good advice.

  • What would be the effect of shikonin on H2O2-induced DNA injury in HL-60 cells?  As an antioxidant, shikonin is expected to attenuate H2O2-induced oxidative injury (Zhong et al DOI: 10.3892/etm.2021.10552). WT1 overexpression is proposed to protect the cells from oxidative stress (Fig. 1E). At the same time, shikonin is supposed to lower the level of WT1 (Fig. 3). How are these two effects compatible with one another?

Response: This is a very good question. However, this question is over this study and is complex. Although shikonin may be an antioxidant, it also promotes cell necroptosis via oxygen-coupled redox cycling (Zhang et al, DOI: 10.1016/j.freeradbiomed.2021.12.314) and can be a broad DNA damage response inhibitor to enhance the anti-cancer effect of chemotherapeutic drugs in both cell cultures and in mouse models(Wang et al, DOI: 10.1016/j.apsb.2021.08.025). This study focused on WT1-associated cell differentiation and did not consider oxidative stress. We suppose that shikonin would abort the protecting effect of WT1 on oxidative stress in HL-60 cells by interfering WT1 function.

 Thanks again for this good question.

Figure 2

These experiments aimed to test correlation between WT1 and CD34 expression (Fig. 2A) and protein-protein interaction of WT1-CD34 (Fig. 2C).

  • 1A: The representative blot does not show significant change in CD34 expression, while a 50% increase is indicated by the relative expression graph. Further independent experiments would be required to strengthen this data.

Response: This is a very useful comment. Based on this comment, we supplied the representative blot which showed significant change in CD34 expression in the revised Fig 2A.

   Thanks again for this good comment.

  • 2B: What is represented by the red lines in the molecular model?

Response: This is a very interesting question. Based on this comment, we explain What is represented by the red lines in the molecular model in the revised Fig 2 legend.

   Thanks again for this good question.

  • 2C: The only experimental proof of WT1-CD34 interaction is this co-IP experiment. The weakness of this experiment it that the IgG control IP sample also reacts with the anti-WT1 antibody. How did you eliminate detection-interference from the heavy-chain (approx. 50kDa) IgG-fragments of the antibody used for the initial immunoprecipitation? Full blots should be shown here to avoid potential misunderstandings. Alternative methods to verify co-localization and interaction of these to proteins would be required (eg. immunofluorescent staining, proximity ligation assay).

Response: This is a very useful comment.  Pre-clearing is not essential when using Protein G-conjugated magnetic beads. However, if the background is high, perform preclearing by incubating 40 µl Protein G-conjugated magnetic beads (50% slurry) and 1 ml diluted lysate (Zhu et al, DOI: 10.3791/55218) . Based on the low background, we did not carry out full blots. However, in order to verify WT1-CD34 interaction, we supplied immunofluorescent staining and added this result in revised Fig2.

   Thanks again for this good comment.

Figure 3

  • 1B: Loading control is of bad quality.

Response: This is a very useful comment. Based on this comment, we replaced Fig. 3B in revised Fig 3.

   Thanks again for this good comment.

  • 1D: How long did you incubate the cells with shikonin before studying the expression of WT1?

Response: This is a very useful question. Based on this comment, we supplied the incubated time in revised Fig.3 legend..

   Thanks again for this good question.

  • What is the supposed mechanism by which shikonin affects both the expression and the protein degradation of WT1?

Response: This is a very useful question.  Nontoxic shikonin as a WT1 inhibitor could decrease CD34 protein level and simultaneously promoted CD11b protein expression in normal HL-60 cells in 48h, whereas WT1 overexpression could reblock the differentiation effect of shikonin on HL-60 cells, indicating a competition relation between WT1 and shikonin in HL-60 cell differentiation.We explained the supposed mechanism  in discussion in the revised manuscript on line 275-280 .

Thanks again for this good question.

 Figure 4

  • 4D: Immunofluorescent images are hardly visible. Better-quality images would be needed to support the quantitative data. Alternatively, mRNA expression or Western blot study of CD34 and CD11b could be applied to verify differentiation.

Response: This is a very useful comment. Based on this comment, we improved Immunofluorescent images in Fig. 4D and supplied western blot study of CD34 and CD11b in the revised manuscript.

   Thanks again for this good advice.

  • line 139:‘the best differentiation effect of 0.4μM shikonin on HL-60 cells was able to be aborted in WT1 overexpressed HL-60O cells’. Could higher doses of shikonin induce differentiation in WT1-overexpressing cells? What concentration of shikonin would be needed to downregulate WT1 in overexpressing cells? These answers would strengthen the idea that shikonin induces HL-60 cell differentiation in a WT1-dependent manner.

Response: This is a very interesting question. This is our mistake im statement. The result that the best differentiation effect of 0.4μM shikonin on HL-60 cells was able to be aborted in WT1 overexpressed HL-60O cells is to verify the interfering effect of shikonin on WT1 function, shikonin downregulated WT1 protein was dependent of dose, higher doses of shikonin was need to induce differentiation in WT1-overexpressing cells. However,  in our pre-experiment, shikonin over 0.4μM could exert cytotoxicity in both HL-60 cells and HL-60O cells, indicating that WT1 overexpression could only reverse  differentiation of shikonin on HL-60 cells.  We discussed this question in discussion on line 275-280 .

   Thanks again for this good question.

Figure 6

  • 6D: What does the green and red color represent during JC-1 staining? Labels should be given next to the images.

Response: This is a very useful comment. Based on this comment, we labeled what  the green and red color represent during JC-1 staining in the revised Fig 6D. We also explain it in figure 6 legend.

   Thanks again for this good advice.

Figure 7

(15) Fig. 7A-F: The color code of Fig. 7D is applied for all graph labelling, but for the other graphs the ‘model’ is labelled with white (instead of black) and ‘shikonin 10 mg/kg’ is labelled with black (instead of blue).

 Response: This is a very useful advice. Based on this comment, we adjusted the color code  in the revised Fig 7.

   Thanks again for this good advice.

  • line 222:‘It was reported that WT1 had a consistent expression or lost with CD34 as cells reverse or differentiate [27].’ This is a confusing sentence. Please, clarify.

Response: This is a very useful comment. Based on this comment, we rewrite this sentence to clearly state what we mean in the revised manuscript.

   Thanks again for this good advice.

  • line 359:The method is described in imperative mode (eg. add, collect, etc), which is unusual in research articles.  Please, modify.

Response: This is a very useful comment. Based on this comment, we redescribed methods and adjusted the declarative mode in the revised manuscript.

   Thanks again for this good advice.

  • line 399 and 411:M-PER protein lysis buffer (not lysate)  

Response: This is a very useful comment. Thanks very much for pointing out this spelling error. We corrected this spelling errors in the revised manuscript.

   Thanks again for pointing out this spelling error.

  • line 413: “Total protein and different concentrations of shikonin … were mixed and incubated at 4° C for 10 h.” Why there is a need for that long incubation?

Response: This is a very useful question. It is our errors for miswriting this incubated time, it is 1h, we corrected this error in the revised manuscript.

   Thanks again for this good question.

  • I have identified some irrelevant references. For example, for WT1 antisense oligonucleotide, reference [23] should be cited. Ref. [40] is about Isothermal Titration Calorimetry, which is absolutely irrelevant to the sentence where it is mentioned (line 259). Isn’t there a shift in reference list? A thorough review of would be necessary.

Response: This is a very useful comment. Based on this comment, we corrected all references and reset them according to required format in the revised manuscript.

   Thanks again for this good advice.

    In addition, we checked all spelling and grammar errors by a native English-speaking colleague. However, limited by specialty and time, what we can do is to clearly state what we mean and minimize the spelling and grammar errors. We hope that you will be satisfied with our revised manuscript

Round 2

Reviewer 2 Report

The authors have answered all the questions and provided better quality figures where it was requested.